# Early identification of preterm neonates at birth with a Tablet App for the Simplified Gestational Age Score (T-SGAS) when ultrasound gestational age dating is unavailable: A validation study

**Archana B. Patel**[1,2]*, **Hemant Kulkarni**[1,3], **Kunal Kurhe**[1], **Amber Prakash**[1], **Savita Bhargav**[1], **Suchita Parepalli**[4], **Elizabeth V. Fogleman**[4], **Janet L. Moore**[4], **Dennis D. Wallace**[4], **Patricia L. Hibberd**[5]

1 Lata Medical Research Foundation, Nagpur, Maharashtra, India, 2 Datta Meghe Institute of Medical Sciences, Sawangi, India, 3 M&H Research, LLC, San Antonio, Texas, United States of America, 4 RTI International, Research Triangle Park, Durham, North Carolina, United States of America, 5 Department of Global Health, Boston University School of Public Health, Boston, Massachusetts Avenue, United States of America

* dr_apatel@yahoo.com

**Data Availability Statement:** The Indian Council of Medical Research does not allow data sharing

## Abstract

### Background

In low resource settings recall of the date of the mother's last menstrual period may be unreliable and due to limited availability of prenatal ultrasound, gestational age of newborns may not be assessed reliably. Preterm babies are at high risk of morbidity and mortality so an alternative strategy is to identify them soon after birth is needed for early referral and management.

### Objective

The objective of this study was to assess the accuracy in assessing prematurity of newborn, over and above birthweight, using a pictorial Simplified Gestational Age Score adapted for use as a Tablet App.

### Methods

Two trained nurse midwives, blinded to each other's assessment and the actual gestational age of the baby used the app to assess gestational age at birth in 3 hospitals based on the following 4 parameters—newborn's posture, skin texture, breast and genital development. Inter-observer variation was evaluated and the optimal scoring cut-off to detect preterm birth was determined. Sensitivity and specificity of gestational age score using the tablet was estimated using combinations of last menstrual period and ultrasound as reference standards to assess preterm birth. The predictive accuracy of the score using the area under a receiver operating characteristic curve was also determined. To account for potential reference standard bias, we also evaluated the score using latent class models.

unless the protocol for data sharing was previously approved by them, the Institutional Ethics Committee (LMRF-IEC) and consented to by the participants. However, other researchers may submit data access requests to Dr. Prabir Kumar Das, Member Secretary at LMRF-IEC (prabir_das23@rediffmail.com).

**Funding:** Grant Number: U10 HF078493 Dr. Archana Patel The study was funded by the Eunice Kennedy Shriver National Institute of Child Health and Human Development's Global Network for Women's and Children's Health Research, USA. While Dr. Kulkarni is affiliated to M&H Research, LLC, San Antonio, TX, USA; this work was done under the aegis of Lata Medical Research Foundation, Nagpur, India of which Dr. Kulkarni is the President and provides his services gratis. No funding was received from M&H Research, LLC for work reported here.The commercial affiliation had no role to play in our study.

**Competing interests:** The authors declare no competing interests.

**Abbreviations:** ANM, Auxiliary Nurse Midwives; CART, Classification And Regression Tree; CCC, concordance correlation coefficient; DS, Dubowitz Score; GA, Gestational Age; IDI, Integrated Discrimination Improvement; LMP, Last Menstrual Period; LR, Likelihood Ratio; LBW, Low Birth Weight; MS, Meharban Singh scale; NBS, New Ballard Score; NRI, Net Reclassification Index; ROC, Receiver Operating Characteristic; T-SGAS, Tablet App for the Simplified Gestational Age Score; USG, Ultrasound.

## Results

A total of 8,591 live singleton births whose gestational age by last menstrual period and ultrasound was within 1 weeks of each other were enrolled. There was strong agreement between assessors (concordance correlation coefficient 0.77 (95% CI 0.76–0.78) and Fleiss' kappa was 0.76 (95% CI 0.76–0.78). The optimal cut-off for the score to predict preterm was 13. Irrespective of the reference standard, the specificity of the score was 90% and sensitivity varied from 40–50% and the predictive accuracy between 74%–79% for the reference standards. The likelihood ratio of a positive score varied between 3.75–4.88 while the same for a negative likelihood ratio consistently varied between 0.57–0.72. Latent class models showed similar results indicating no reference standard bias.

## Conclusion

Gestational age scores had strong inter-observer agreement, robust prediction of preterm births simplicity of use by nurse midwives and can be a useful tool in resource-limited scenarios.

## Trial registration

The Tablet App for the Simplified Gestational Age Score (T-SGAS) study was registered at ClinicalTrials.gov NCT02408783;

## Introduction

Globally, 15 million babies are born preterm every year [1]. These babies are difficult to identify at birth, particularly in resource limited settings where gestational age (GA) may not be accurately assessed prenatally by ultrasound (USG) and estimates of the traditionally used method of estimating GA using the first day of her last menstrual period (LMP) may be unreliable. Since preterm babies are at high risk of morbidity and mortality, it is necessary to estimate GA at birth for early referral and management.

The estimation of GA using LMP is dependent on the regularity of the pregnant woman's menstrual cycle, as well as an accurate recall by her of the date of the first day of the last period. It has been reported that this date is uncertain in 20% of pregnant women [2]. Theoretically, if a woman enrolls early for prenatal care and has a USG then the early USG can be used for ascertainment of GA, especially if LMP is unreliable. However, women from LMICs usually enroll late for prenatal care and are less likely to remember the date of LMP. Further, there is a paucity of USG availability in these countries and it is not available at rural community health facilities. In facilities where USG is available it is prioritized for women in the third trimester of pregnancy. USG is less predictive of GA when estimated late in pregnancy [3, 4]. Therefore, newborns are often identified as preterm just after birth by community birth attendants at the community health facilities. In such circumstances, accurate and feasible methods of determining postnatal GA by the community birth attendants would help early recognition and facilitate referral of preterm newborns for appropriate management.

Postnatal GA assessment scores such as Dubowitz Score (DS) and New Ballard Score (NBS) have been used widely but are complex and require neurological examination, and may be altered when a newborn has morbidities like birth asphyxia, infection or congenital anomalies. Another limitation is that these scores have not been evaluated for regional, geographical,

racial or ethnic robustness. The Meharban Singh scale (MS) was developed in India in 1975. Although it is much simpler than the NBS and the DS, it still has 11 items to score on and not suitable for use by community birth attendants or Auxiliary Nurse Midwives (ANMs) in India, who deliver newborns at rural health facilities. For these reasons, there is a need of a simple tool for use by ANMs to assess GA at birth and to make appropriate referral decisions [5–7]. Considering these implementation gaps, we developed a tablet-based pictorial GA assessment tool, the "Tablet based—Simplified Gestational Age Score" or T-SGAS. T-SGAS was adapted from a previously developed and tested simplified gestational age scoring system [8, 9]. The aim of this study was to assess the accuracy of T-SGAS in assessing prematurity of newborn, over and above birthweight.

## Methods

### Overall study design

This was a cross-sectional validation study to evaluate the accuracy of T-SGAS to ascertain postnatal GA within 24 hours of birth. Details of the methods used have been described elsewhere [8]. Briefly, the study was conducted at three second-level care birthing facilities in and around Nagpur in central India. The three study sites in India were: 1) Daga Memorial Women's Hospital, Nagpur, 2) General Hospital, Bhandara and 3) General Hospital, Wardha. These facilities were selected to choose a mixture of rural as well as urban populations, to obtain samples from all GA categories and to get a good representative sample for normal/uncomplicated as well as complicated deliveries. All three hospitals are well equipped for maternal and child-care facilities including labor rooms, caesarean section facility, intensive care units, blood banks and neonatal care units. These facilities serve as district-level referral facilities, providing routine care services under the state public health system.

The mothers of all singleton live births were consented for assessment of the GA of their newborns within 24 hours of birth. Real-time data were collected on tablet-based forms by trained ANMs. T-SGAS is a pictorial App that consisted of photographs that were the best fit for each score of the following four items: newborn's posture (score 0 to +4), skin (score -1 to +5), breast (score -1 to +4) and genitals (score -1 to +4). Fig 1 The T-SGAS app then auto-calculated the total score and classified the newborn's GA category. If the total score was <14, the newborn was considered <37 weeks and classified as preterm. The forms also included date of delivery, time of delivery, delivery outcome, gender of newborn and consent status. Another form on the tablet collected data on maternal and newborn demographics, birth weight, LMP, details of USG, and assessment for the study eligibility. This data was collected by an independent data collector from the mother's hospital record.

### Participant selection and recruitment of newborns for the validation of T-SGAS

Each facility had a team of 2 ANMs (ANM1 and ANM2) and a data collector working in two, twelve-hour shifts. All mothers above the age of 18 years (legal consenting age), with singleton live births, with a known LMP and with a report of at least one prenatal ultrasound assessment of GA (if more than one, then the earliest ultrasound GA was recorded) gave written informed consent for GA evaluation of their newborn. ANM1 was responsible for this initial screening and consent of the mother. ANM2 was responsible for assessing the same newborn, independently, within 24 hours of birth and after ANM1 had completed and saved her assessment. Once the assessment was completed the data form on the tablet was closed and could not be accessed again. So, ANM1 and ANM2 were blinded to each other's GA assessment scores. Both

| Maturity Sign | Score | | | | | | |
|---|---|---|---|---|---|---|---|
| | -1 | 0 | +1 | +2 | +3 | +4 | +5 |
| Posture | | Upper as well as lower limbs completely extended | Overall posture extended, some flexion at ankles and knees | Knees are well flexed, with slight flexion at elbows or hips | Hips are well flexed but without abduction | Hip abduction accompanies flexion (acute angles at the hips) | |
| Skin | Sticky, Friable, Transparent | Gelatenous, Red, Translucent | Smooth pink, Visible veins | Superficial peeling +/- Rash, Few veins | Cracking, Pale areas, Rare veins | Parchment, Deep cracking, No vessels | Leathery, Cracked, Wrinked |
| Breast | Imperceptible | Barely perceptable | Flat areola, No bud | Stippled areola, 1-2 mm bud | Raised areola, 3-4 mm bud | Full areola, 5-10 mm bud | |
| Genital Male | Scrotum flat, smooth | Scrotum empty, faint rugae | Testes in upper canal, rare rugae | Testes descending, few rugae | Testes down, good rugae | Testes pendulous, deep rugae | |
| Genital Female | Clitoris prominent and labia flat | Prominent clitoris and small labia minora | Prominent clitoris and enlarging minora | Majora and minora equally prominent | Majora large, minoral small | Majora covers clitoris and minora | |

**Fig 1. Pictorial representation of the neonatal characteristics for the Tablet App for the Simplified Gestational Age Scoring System (T-SGAS).**

ANMs were blinded to the neonates' actual GA based on LMP or USG that was recorded separately by a data collector from the mother's hospital record and the delivery chart. The data collector ascertained the neonate's data eligibility for analysis after the ANMs had finished assessing the neonate. This was done to avoid any possibility of the ANM's knowing the GA of the neonate. Data collection and transmission was on real-time basis to a central data server [8].

## Statistical analysis

Analyses were conducted in Stata 13.1 (Stata Corp, College Station, TX), the R package TREE [10] (for classification tree analysis) and the LATENT1 program (Latent1 Software, Version 3, McMaster University, Hamilton, Ontario, Canada, for latent class analyses). Statistical

significance was tested at a type I error rate of 0.05. When multiple tests were conducted an appropriate Bonferroni correction [11] was applied to adjust the type I error rate to a global value of 0.05. All the analyses were conducted in two subsets of the data: those in whom the LMP and USG estimates of GA were within one week of each other (n = 8,591) and within two weeks of each other (n = 11,305). Agreement between different T-SGAS assessments was analyzed in three steps. Firstly, we used the total T-SGAS scores from the two assessments and estimated Lin's concordance correlation coefficient [12] (CCC). Secondly, to test the strength of agreement, as the paired assessments were done by several combinations of ANMs, we used the difference in these two scores to estimate Fleiss's kappa [13] to complement the CCC estimates. Thirdly, we used the dichotomous categorizations of preterm obtained from the two assessments and estimated Fleiss's kappa on this dichotomous classification. Our previous studies have shown that a score of 14 or more is indicative of a term baby [9]. This cut-off was used to generate the dichotomous categories from the two assessments.

Screening accuracy of the T-SGAS assessments was determined using two complementary approaches. First, a preterm birth was defined when GA at birth was <37 weeks by the following four reference standard combinations: LMP alone, USG alone, either LMP or USG and both LMP and USG. Using each of these reference standards for classifying a birth as preterm, we first estimated the prevalence of preterm birth and the sensitivity and specificity of T-SGAS assessments. We also determined the predictive accuracy using the area under a receiver operating characteristic (ROC) curve. The ROC curves were based on the predicted probability estimates obtained from logistic regression models. We evaluated the effect of potential confounding on the performance of T-SGAS by the enrolling facility (as a reflection of the influence of prevalence on screening performance), gender of the newborn (since one of the components of T-SGAS is based on the maturity of genitalia) and maternal age (as a reflection of varying likelihood of a preterm birth). Second, since none of these four assessments (ANM1 and ANM2 measurements from T-SGAS and the LMP and USG measurements) can be considered as a 'true' reference standard, we used Walter and Hui's latent class models to estimate the prevalence of a preterm birth and the sensitivity and specificity of the ANM1 and ANM2 assessments.

Classification and regression tree (CART) analysis was used to determine the optimal cut-off for dichotomization of the T-SGAS score which was data-driven from this data set. Birth weight and the T-SGAS assessments were used as predictors in these analysis to identify clinically meaningful risk groups. Also, each of the four aforementioned reference standards (LMP alone, USG alone, either LMP or USG, and, both LMP and USG) was separately considered as the dependent variable and birth weight and total T-SGAS scores were considered as the independent variables. Incremental values of ANM1 and ANM2 assessments over birth weight as a predictor of preterm birth was determined using integrated discrimination improvement (IDI) and net reclassification index (NRI) [14].

### Ethics statement

The study protocol and the written informed consent documents were reviewed and approved by the Institutional Review Boards (IRBs) and ethics committees of the Lata Medical Research Foundation IRB (FWA00012971), the Partners Human Research Committee, Boston, MA, and the Boston University Medical Campus IRB.

## Results

### Study participants

Fig 2 shows the study flow diagram. Of the 15,920, live births enrolled into the study a total of 8,591 had estimates of GA by LMP and GA by USG within one week of each other and a total

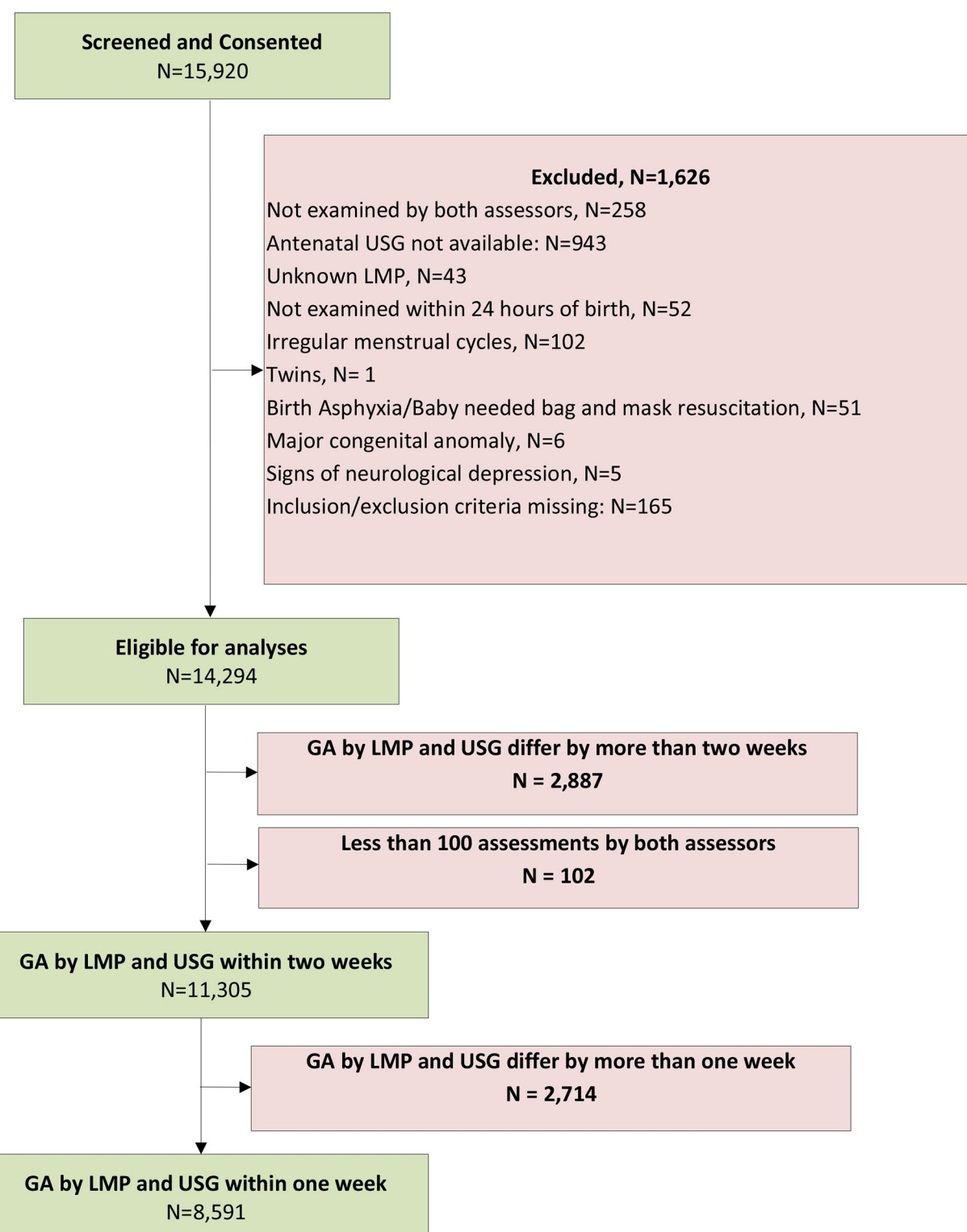

**Fig 2. SGAS consort diagram showing the population of n = 8591 women with singleton birth included in the analysis.**

of 11,305 (71%) live births had GA by LMP and USG within 2 weeks of each other. Characteristics of the mother and newborn were similar in the included larger (n = 11,305) when compared to the excluded (n = 4,615).The average maternal age of the included live births at conception was 24 years, ~93% of the mothers had received at least a secondary level of education, the average parity was one, and in a majority of cases the mode of delivery was either vaginal or caesarean section. However, the proportion of LBW babies was significantly larger in the excluded babies as compared to the included babies (30.0% versus 18.9%, p <0.001). The average GA at USG was 32.1±5.9 weeks for the larger dataset and 31.7±6.2 weeks in the subset of 8,591. For simplicity of presentation, henceforth we report the results from the more constrained data set (n = 8,591) as the primary results. Similar analyses (tables and figures) for the full set of 11,305 live births are shown in S1 Table.

## Interobserver agreement of T-SGAS scores

Using the Lin's concordance correlation coefficient, the agreement in the scores between two assessors (ANMs) was 0.77 (95% CI 0.76–0.78). Table 1 shows an excellent agreement among the assessors (Fleiss's kappa = 0.76, 95% CI 0.75–0.78) with 88.3% of the scores were within zero or one point. We also studied the agreements between the assessments (rather than the assessors) classifying as preterm or term, Fleiss's kappa was still high (0.69, 95% CI 0.67–0.70) with 85.5% assessments in complete agreement with each other.

## Accuracy of the T-SGAS to predict gestational age

We tested the hypothesis that lower T-SGAS scores are associated with a higher likelihood of a preterm birth. Given the strong agreement between the T-SGAS scores obtained by two assessments, we conducted these analyses using the average T-SGAS score as the predictor variable. Logistic regression analyses using the four reference standards for assessing preterm (based on LMP alone, USG alone, LMP or USG, and LMP and USG) consistently indicated that the probability of a preterm birth was high when scores were below 10, then rapidly declined for scores between 11 and 13 and remained low thereafter (Fig 3A). ROC curves (Fig 3B) showed that the predictive accuracy of T-SGAS varied between 74%– 79% for the four reference standards for assessing GA.

## Risk stratification using birth weight and T-SGAS

To identify a critical cut-off point of T-SGAS score for identification of a preterm birth, we conducted a series of CART analyses. Results for the first assessor using LMP as a reference

**Table 1. Agreement between T-SGAS scores by two assessors (analysis subset: LMP and USG estimates of GA within one week of each other, n = 8,591).**

| Characteristic | Description |
|---|---|
| Agreement between assessors | |
| Complete agreement [n (%)] | 5,150 (59.9) |
| 1 point [n (%)] | 2,438 (28.4) |
| 2 points [n (%)] | 703 (8.2) |
| 3 points [n (%)] | 199 (2.3) |
| 4+ points [n (%)] | 101 (1.2) |
| Fleiss's kappa (95% CI) | 0.76 (0.75–0.78) |
| Agreement between assessments | |
| Complete agreement [n (%)] | 7,349 (85.5) |
| Fleiss's kappa (95% CI) | 0.69 (0.67–0.70) |

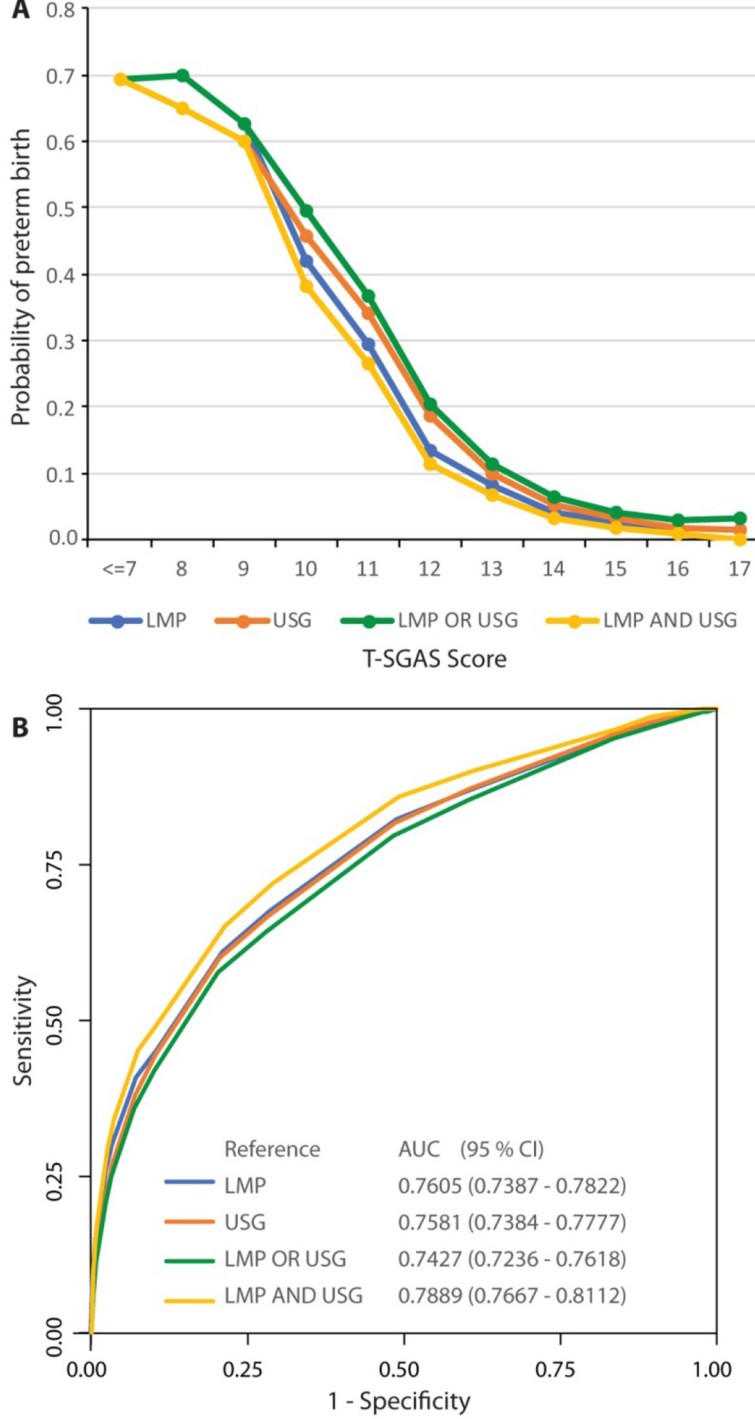

**Fig 3. Accuracy of T-SGAS score to predict gestational age and preterm birth in n = 8,591 participants.**

standard are shown in Fig 4 and indicated that the overall probability of a preterm birth was ~7%. A combination of birth weight and T-SGAS scores created four groups with widely varying probability profiles. Those with a CART cut-off of birth weight < 2.575 kg and T-SGAS score <10 had a 65% probability of preterm birth. Those with birth weight < 2.575 kg but a

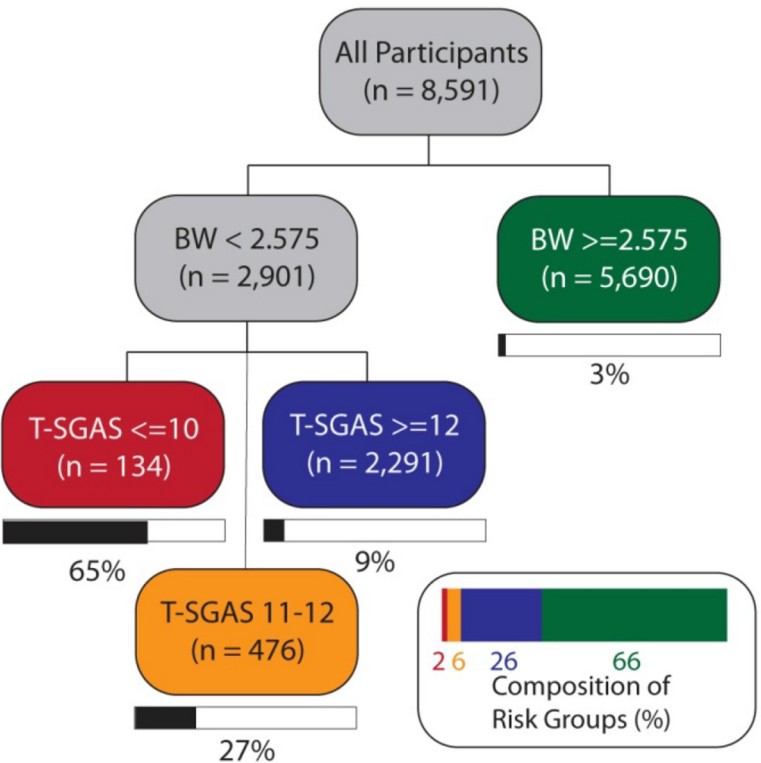

**Fig 4. Classification tree analysis for the outcome of LMP in n = 8,591 participants.**

T-SGAS score of 11–12 had a 27% probability of preterm birth and those with birth weight < 2.575 kg but a T-SGAS score of 13 or more had a 9% probability of preterm birth. In contrast, those with a birthweight exceeding 2.575 kg had a probability of preterm birth at 3%. This cut-off of a T-SGAS score of 13 or more as an indicator of a term birth was consistently seen for all the four reference standards (S1A–S1C Fig). These analyses were also consistent for the second assessor results. Therefore, for all ensuing analyses we used a T-SGAS score <13 as indicative of a preterm birth.

### Screening performance of the T-SGAS to identify a preterm birth

Using the cut point of score of 13, we determined the accuracy of T-SGAS to identify a preterm birth. This set of analyses was done in two steps. In the first step, we used each of the four reference standards (LMP alone, USG alone, LMP or USG, LMP and USG) to directly compare the screening performance of both T-SGAS assessments. These results are shown in Table 2. The prevalence of a preterm birth varied between 5.6% to 9.5% based on the reference standard used. Our analyses indicated that irrespective of the reference standard the specificity (ability to detect a 'true' term baby) of T-SGAS score remained high (~90%). The sensitivity of T-SGAS varied between 40% to 50%. The sensitivity estimates were sensitive to the definition of a preterm birth such that when GA was estimated to be <37 weeks by both LMP and USG (column titled LMP AND USG in Table 2 and S2 Table), the sensitivity of T-SGAS score was highest and reached close to 50%. The likelihood ratio of a positive T-SGAS (LR+) remained high, irrespective of the reference standard used or the dataset analyzed, and varied between 3.75–4.88. Similarly, the LR of a negative T-SGAS (LR-) also was consistently observed to be 0.57–0.72. Potential confounding on the performance of T-SGAS by the enrolling facility,

**Table 2. Screening accuracy of T-SGAS to predict a preterm birth (analysis subset: LMP and USG estimates of GA within one week of each other, n = 8,591).**

| Parameter | Reference Standard | | | |
|---|---|---|---|---|
| | **Last Menstrual Period (LMP)** | **Ultrasonography (USG)** | **LMP OR USG** | **LMP AND USG** |
| | *Assessor 1* | | | |
| Prevalence | 6.83 (6.31–7.39) | 8.25 (7.68–8.85) | 9.48 (8.86–10.11) | 5.61 (5.13–6.12) |
| Sensitivity | 43.27 (39.22–47.39) | 42.45 (38.78–46.19) | 40.05 (36.66–43.51) | 47.51 (42.98–52.08) |
| Specificity | 90.07 (89.39–90.71) | 90.51 (89.84–91.15) | 90.70 (90.04–91.34) | 89.89 (89.21–90.54) |
| Positive Predictive Value | 24.21 (21.65–26.92) | 28.69 (25.97–31.54) | 31.08 (28.29–33.98) | 21.83 (19.36–24.46) |
| Negative Predictive Value | 95.58 (95.10–96.04) | 94.59 (94.06–95.09) | 93.53 (92.95–94.07) | 96.65 (96.21–97.04) |
| Likelihood Ratio + | 4.36 (3.89–4.88) | 4.47 (4.01–4.99) | 4.31 (3.86–4.80) | 4.70 (4.19–5.27) |
| Likelihood Ratio - | 0.63 (0.59–0.68) | 0.64 (0.60–0.68) | 0.66 (0.62–0.70) | 0.58 (0.54–0.64) |
| | *Assessor 2* | | | |
| Prevalence | 6.83 (6.31–7.39) | 8.25 (7.68–8.85) | 9.48 (8.86–10.11) | 5.61 (5.13–6.12) |
| Sensitivity | 44.63 (40.56–48.76) | 41.47 (37.81–45.19) | 39.43 (36.06–42.89) | 48.76 (44.21–53.32) |
| Specificity | 90.22 (89.55–90.86) | 90.47 (89.80–91.11) | 90.69 (90.02–91.33) | 90.01 (89.34–90.66) |
| Positive Predictive Value | 25.07 (22.47–27.82) | 28.13 (25.42–30.97) | 30.72 (27.93–33.61) | 22.49 (19.99–25.14) |
| Negative Predictive Value | 95.69 (95.21–96.14) | 94.50 (93.96–95.00) | 93.47 (92.89–94.01) | 96.73 (96.30–97.12) |
| Likelihood Ratio + | 4.56 (4.08–5.10) | 4.35 (3.90–4.86) | 4.24 (3.80–4.73) | 4.88 (4.36–5.46) |
| Likelihood Ratio - | 0.61 (0.57–0.66) | 0.65 (0.61–0.69) | 0.67 (0.63–0.71) | 0.57 (0.52–0.62) |

gender of the newborn and maternal age had no effect (S3, S4 and S5 Tables, respectively). This interpretation could be consistently made for all combinations of assessors, reference standards and datasets used for analyses. We then ran latent class analyses for assessing preterm using T-SGAS score, LMP and USG as three independent tests, none of which were considered as a reference standard. Our results (S6A and S6B Table) were consistent with the results obtained in the previous step. These results showed that the estimated prevalence of preterm births was ~8%. Also, all three methods have very high specificity (90% or above) but the sensitivity of USG was the highest (above 90%), followed by LMP (~80%) and then by T-SGAS (~50%). Using these results, we obtained the estimates of LR of a positive T-SGAS. We found that after accounting for possible classification errors associated with imperfect LMP and USG through latent class analyses, the LR+ of T-SGAS varied between 5.14 to 5.27 (95% CI varying from 4.63–5.85), indicating a strong and significant predictive value of T-SGAS.

### Incremental value of T-SGAS over birth weight

We conducted these analyses for both assessors and all four reference standards mentioned earlier. As indicated by the integrated discrimination improvement (IDI) we observed that T-SGAS improved the discrimination by 5–7% irrespective of the assessor. And similarly, the net reclassification index (NRI) estimates demonstrated an improved reclassification when T-SGAS were used in addition to birth weight for classification of live births as preterm. (S7A and S7B Table)

All of the above described analyses were run on the full set of 11,305 live births and concurring results were obtained which are shown as S2–S9 Tables and S1–S4 Figs.

### Discussion

Our main findings were that there was strong agreement between the ANM assessors (concordance correlation coefficient 0.77 (95% CI 0.76–0.78) and Fleiss' kappa was 0.76 (95% CI 0.76–0.78)). The optimal cut-off for the score to predict preterm was 13 and irrespective of the

reference standard used (LMP, USG, LMP Or USG, LMP and USG) the specificity of the score was 90% and sensitivity varied from 40–50% and the predictive accuracy between 74%–79% for the reference standards. Latent class models showed similar results indicating no reference standard bias.

A systematic review of 18 newborn assessments for GA estimation (ranging from 4 to 23 signs) concluded that assessments with fewer signs tended to be less accurate. There was a tendency for newborn assessments to overestimate GA in preterm and underestimate GA in growth-restricted infants [15]. They recommended that where USG is not available simple yet specific approaches be developed. Our T-SGAS was different from all previously used GA assessments methods that are not only complex but rely on rigorous training of health workers who then have to recall the description of the items to match the score during actual assessments. T-SGAS is simple with just a simple 4 item scoring scale with 3 physical and 1 neurological item. T-SGAS was developed by using the most predictive items from the 12-item New Ballard Score (NBS), 22-item Dubowitz score (DWS) and 11-item Meharban Singh (MS) and validated against the Best obstetrics Estimate (BOE) in low birth weight (LBW) newborns in India [9]. Because it showed promising results for estimation of GA in LBW newborn and was easy to by nursing staff in the community it was adapted to a pictorial tablet app (T-SGAS) [8]. In the current study the ANMs were trained to use only the T-SGAs and it was validated against the gold standard of GA at birth assessed using prenatal USG, which is also the gold standard for validating any newborn gestational age assessment methods like NBS or DWS. So, validation against NBS or DWS was not considered necessary as any additional training provided to ANMs on NBA or DWS would bias the evaluation skills of ANMs. T-SGAS is unique because it is a pictorial scoring App that contains real life newborn pictures of all the items that match each score. These pictures were obtained from the local preterm population and carefully selected by concurrence between 3 neonatologists to exactly match the description of the scores. The ANMs have to simply select the picture that matches what they observe in the newborn baby. The App then auto-calculates the score and categorizes the newborn as preterm. Therefore, it is not surprising that the assessments were reliable and the concordance between two assessors was high.

The LR+ of T-SGAS was sufficiently high (~5) to warrant its routine use. Using Pauker-Kassirer approach, we estimated that even if the benefit-to-risk ratio of early detection to a missed diagnosis of a PT birth was as low as 5, the testing threshold to use T-SGAS would be as little as 0.0385. This indicates that if the prevalence of preterm births exceeds 3.85%, there will be a screening value to using the T-SGAS. In developing countries like India, the prevalence of a preterm birth is likely to be high and thus T-SGAS use would be highly relevant in such situations. Furthermore, the benefit-to-risk ratio of early detection of a preterm birth is likely to be much higher in reality, making the routine use of T-SGAS fully justifiable. It is noteworthy that the T-SGAS scores were not significantly influenced by observer variability, maternal age, gender of the newborn and the prevalence differential across enrolling institutions. These observations lend credence to the robustness of T-SGAS.

We noted a relatively low sensitivity (~50%) of the T-SGAS scores to predict a preterm birth. Of note, sensitivity and specificity–while theoretically independent of the prevalence– may vary with prevalence [16]. Also, from a clinical point-of-view, likelihood ratios are superior to sensitivity and specificity by being prevalence-agnostic as well as predictive of the disease condition [17]. Our estimates of LR+ for T-SGAS far outweigh the relatively low sensitivity observed in this study. When we aimed to increase the sensitivity by moving the T-SGAS score cut-off, the LR+ estimates were smaller indicating a substantial loss in specificity. For these reasons, we persisted with the cut-off of a total score of 13 as the most discriminatory one.

Conceptually, gestational age–like any age variable–should be measured on the time scale. Thus, use of LMP is a natural choice for GA measurement but the heavy recall bias associated with LMP places demands for alternative measures of GA. USG measures the fetus size and correlates the size with GA, but there are three important constraints for USG use in resource-limited settings: relatively expensive, not always available and inaccurate when USG assessment is done later in pregnancy. As observed in our study (S2 Fig), women in developing countries tend to get USG assessments late during pregnancy. Our proposed method (T-SGAS) circumvents these issues associated with LMP and USG and is based on maturity indicators that are simple enough to be identified by an ANM but robust enough to be routinely used.

Two limitations of our study must be recognized. First, this study is of a cross-sectional nature in sub-population of Indian babies from Central India and the prognostic value of the risk groups on neonatal or early childhood outcomes such as morbidity and mortality are unknown. Future studies need to assess the influence of T-SGAS-based risk-stratification on hard clinical endpoints. Second, the primary aim of this study was to validate the accuracy of the T-SGAS score, but prudent ways to combine LMP (when reliable) and USG (when available) need to be devised to further improve the prediction of a preterm birth. Future validation studies from other parts of India are needed before it be recommended for integration into the existing health care services.

Notwithstanding these limitations, our results clearly demonstrate the value of T-SGAS in risk-stratification for identification of preterm births. Interestingly, by redefining the threshold used (37 weeks) in this study, it is conceivable to address other clinically important questions using our tool. For example, if a threshold of 35 weeks is used then the need for admission to a neonatal intensive care unit may also be predicted by this tool. Future studies are needed to directly address such interesting questions. In conclusion, T-SGAS demonstrates strengths of high interobserver agreement, robust and accurate prediction of preterm births, incremental value over birth weight and simplicity of use and interpretation. We advocate strongly in favour of further validation and use of T-SGAS for GA assessment and PT identification, especially in resource-limited scenarios.

To conclude, in the context of preterm births, there exists a significant gap between needs and resources. Developing countries that have among the highest prevalence of preterm births have least availability of the required health facilities for prevention and management of this scourge [18]. It is therefore important and imperative to optimize the available resources by adequate risk-stratification. In peripheral settings where expert medical care is not readily available, it is conceivable that local resources like the ANMs can be trained to identify preterm births thereby providing early referral services to the needy. Our results support this contention and indicate that a simple and inexpensive method of GA assessment can be used in the periphery to optimize health resource use for management of a preterm birth.

## Supporting information

**S1 Fig. CART analyses for other outcomes for Assessor 1.**
(TIF)

**S2 Fig. Cumulative proportion of women who had USG assessment till the gestational age at the time of USG.**
(TIF)

**S3 Fig. Accuracy of T-SGAS score to predict gestational age and preterm birth in the full set of 11,305 participants.**
(TIF)

**S4 Fig. Classification tree analysis for the outcome of LMP in the full set of 11,305 participants corresponding to Fig 3 (main text).**
(TIF)

**S1 Table. Equivalence between included and excluded participants.**
(DOCX)

**S2 Table. Screening accuracy of T-SGAS to identify preterm births when estimates of GA by LMP and USG were within 2 weeks of each other (n = 11,305).**
(DOCX)

**S3 Table. Stratified Mantel-Haenszel analyses to investigate the potential influence of enrolling institutions on estimates of T-SGAS accuracy.**
(DOCX)

**S4 Table. Stratified Mantel-Haenszel analyses to investigate the potential influence of the gender of newborn on estimates of T-SGAS accuracy.**
(DOCX)

**S5 Table. Stratified Mantel-Haenszel analyses to investigate the potential influence of maternal age quartiles on estimates of T-SGAS accuracy.**
(DOCX)

**S6 Table.** a. Results of latent class analyses (analysis subset: LMP and USG estimates of GA within one week of each other, n = 8,591). b. Results of latent class analyses when estimates of GA by LMP and USG were within 2 weeks of each other (n = 11,305).
(DOCX)

**S7 Table.** a. Incremental value of T-SGAS over birth weight (analysis subset: LMP and USG estimates of GA within one week of each other, n = 8,591). b. Incremental value of T-SGAS over birth weight when estimates of GA by LMP and USG were within 2 weeks of each other (n = 11,305).
(DOCX)

**S8 Table. Combinations of ANM assessors who evaluated the live births.**
(DOCX)

**S9 Table. Agreement between assessor pairs when estimates of GA by LMP and USG were within 2 weeks of each other (n = 11,305).**
(DOCX)

## Acknowledgments

We thank Dr. Akash Bang, Dr. Ashish Lothe, Dr. Archana Patel, Dr. Mohini Apte, Dr. Vinita Jain, Dr. Satish Deopujari, Dr. Kuldeep Sukhdeve, the ANMs, all the hospitals, the mothers and babies that participated in the successful conduct of this study.

## Author Contributions

**Conceptualization:** Archana B. Patel, Kunal Kurhe, Amber Prakash, Suchita Parepalli, Patricia L. Hibberd.

**Formal analysis:** Hemant Kulkarni, Amber Prakash, Elizabeth V. Fogleman, Janet L. Moore, Dennis D. Wallace.

**Investigation:** Archana B. Patel.

**Project administration:** Kunal Kurhe, Savita Bhargav, Patricia L. Hibberd.

**Software:** Kunal Kurhe, Amber Prakash, Suchita Parepalli.

**Supervision:** Savita Bhargav, Patricia L. Hibberd.

**Validation:** Suchita Parepalli.

**Writing – original draft:** Archana B. Patel, Hemant Kulkarni.

**Writing – review & editing:** Archana B. Patel, Hemant Kulkarni, Kunal Kurhe, Amber Prakash, Savita Bhargav, Suchita Parepalli, Elizabeth V. Fogleman, Janet L. Moore, Dennis D. Wallace, Patricia L. Hibberd.

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
