## [Decision Letter · Decision Letter 0]

5 May 2020

PONE-D-20-09295

Early identification of preterm neonates at birth with a Tablet App for the Simplified Gestational Age Score (T-SGAS) when ultrasound gestational age dating is unavailable: A validation study

PLOS ONE

Dear Dr. Patel,

Thank you for submitting your manuscript to PLOS ONE. After careful consideration, we feel that it has merit but does not fully meet PLOS ONE’s publication criteria as it currently stands. Therefore, we invite you to submit a revised version of the manuscript that addresses the points raised during the review process.

We would appreciate receiving your revised manuscript by Jun 19 2020 11:59PM. To enhance the reproducibility of your results, we recommend that if applicable you deposit your laboratory protocols in protocols.io, where a protocol can be assigned its own identifier (DOI) such that it can be cited independently in the future. For instructions see: http://journals.plos.org/plosone/s/submission-guidelines#loc-laboratory-protocols

We look forward to receiving your revised manuscript.

Kind regards,

Prem Singh Shekhawat, MD

Academic Editor

PLOS ONE

Additional Editor Comments (if provided):

The submission titled "Early identification of preterm neonates at birth with a Tablet App for the Simplified Gestational Age Score (T-SGAS) when ultrasound gestational age dating is unavailable: A validation study" by Archana Patel et al is an attempt to come up with a screening tool to estimate gestational age of neonates in a resource poor setting so that special attention may be given to infants born <37 weeks to prevent morbidity and mortality. Authors report on use of a android tablet app by two trained midwife's in 4 community hospitals in India to assess gestational age with an aim to designate each infant as either <37 weeks or >37 weeks. This tool becomes highly relevant in countries like India where majority of deliveries are conducted by nurse midwife's and an assessment of GA helps screen neonates who need special attention and referral to an appropriate tertiary care center. This is a well-designed study and a well-written manuscript and has the potential for widespread use in the third world countries. I would like authors to make several major and minor changes to their submission before it can be considered suitable for publication.

Major concerns:

1. T-SGAS a pictorial App that consisted of photographs to score four items: newborn’s posture (score 0 to +4), skin (score - 134 1 to +5), breast (score -1 to +4) and genitals (score -1 to +4), seems very interesting compared to other established tools like Dubowitz or Modified Ballard. It will be very helpful to our readers if authors could add a figure to the manuscript to depict the images used to do this said scoring. In its current form this manuscript does not provide a visual impression of the assessment being made to our readers.

2. Was the GA assessment also done by established norms like Dubowitz or Ballard's methods by a trained professional? If yes, then how these assessments varied compared to the app? If not then why not? Kindly explain it in the discussion part of the manuscript.

3. Discussion needs to elaborate on how this tool can be used by ANM's and in what way it can be helpful to these designated infants <37 weeks?

4. Table 3 and 4 though helpful to a trained professional with good knowledge of biostatistics, does not help our average reader of PLOSone so should be either deleted or published as supplementary tables on the web.

Minor concerns:

1. Remove repetitive statements in introduction, methods and results to reduce the size of manuscript.

2. Elaborate on educational status on ANM's in your methods section and why they were chosen to do the measurements, were they randomly selected or had special qualifications?

2. Please provide additional details regarding participant consent. In the Methods section, please ensure that you have specified what type of consent you obtained (for instance, written or verbal) and whether the ethics committee approved this consent procedure. If verbal consent was obtained please state why it was not possible to obtain written consent and how verbal consent was recorded. If your study included minors, state whether you obtained consent from parents or guardians.

"The authors declare no competing interests."

We note that one or more of the authors are employed by a commercial company: M&H Research, LLC

7. Your ethics statement must appear in the Methods section of your manuscript. If your ethics statement is written in any section besides the Methods, please move it to the Methods section and delete it from any other section. Please also ensure that your ethics statement is included in your manuscript, as the ethics section of your online submission will not be published alongside your manuscript.

Reviewers' comments:

Reviewer's Responses to Questions

**Comments to the Author**

1. Is the manuscript technically sound, and do the data support the conclusions?

Reviewer #1: Yes

2. Has the statistical analysis been performed appropriately and rigorously? 

Reviewer #1: Yes

3. Have the authors made all data underlying the findings in their manuscript fully available?

Reviewer #1: Yes

4. Is the manuscript presented in an intelligible fashion and written in standard English?

Reviewer #1: Yes

5. Review Comments to the Author

Reviewer #1: This is a cross-sectional study evaluating the accuracy of Tablet based gestational age estimation app in low resource setting. They have identified score <14 as good predictor of identifying preterm birth (<37 weeks) and 9% probability of preterm birth if score>13 and weight <2.5kg.

The app has high Inter & intra-observer variation and Likelihood ratio, though sensitivity is low. The study is well designed and the analysis is robust and potential for being a useful tool in low-resource settings.

Concerns:

1. 37 weeks was used as the cutoff for identifying preterm birth or not. How valid is the tool at other gestational ages - <32 weeks, <35 weeks etc. <37 weeks is important for epidemiological reasons to identify preterm birth. However, infants >35 weeks generally don’t require admission to NICU. This data would be more helpful for clinicians whether the baby needs to be transported to the NICU or not, which in a low resource NICU could be a significant cost.

2. Can you clarify if the ANM assessor’s correlation coefficient was for assessors from all three study sites?

3. Likelihood ratios should be included in the abstract.

6. PLOS authors have the option to publish the peer review history of their article (what does this mean?). If published, this will include your full peer review and any attached files.

Reviewer #1: No

---

## [Author Response · Author response to Decision Letter 0]

28 Jul 2020

• A rebuttal letter that responds to each point raised by the academic editor and reviewer(s). This letter should be uploaded as separate file and labeled 'Response to Reviewers'.

• A marked-up copy of your manuscript that highlights changes made to the original version. This file should be uploaded as separate file and labeled 'Revised Manuscript with Track Changes'.

• An unmarked version of your revised paper without tracked changes. This file should be uploaded as separate file and labeled 'Manuscript'.

RESPONSE TO REVIEWER

Additional Editor Comments (if provided):

The submission titled "Early identification of preterm neonates at birth with a Tablet App for the Simplified Gestational Age Score (T-SGAS) when ultrasound gestational age dating is unavailable: A validation study" by Archana Patel et al is an attempt to come up with a screening tool to estimate gestational age of neonates in a resource poor setting so that special attention may be given to infants born <37 weeks to prevent morbidity and mortality. Authors report on use of a android tablet app by two trained midwife's in 4 community hospitals in India to assess gestational age with an aim to designate each infant as either <37 weeks or >37 weeks. This tool becomes highly relevant in countries like India where majority of deliveries are conducted by nurse midwife's and an assessment of GA helps screen neonates who need special attention and referral to an appropriate tertiary care centre. This is a well-designed study and a well-written manuscript and has the potential for widespread use in the third world countries. I would like authors to make several major and minor changes to their submission before it can be considered suitable for publication.

Response: We are thankful to you for careful review and understanding the relevance of this paper in India.

Major concerns:

1. T-SGAS a pictorial App that consisted of photographs to score four items: newborn’s posture (score 0 to +4), skin (score - 134 1 to +5), breast (score -1 to +4) and genitals (score -1 to +4), seems very interesting compared to other established tools like Dubowitz or Modified Ballard. It will be very helpful to our readers if authors could add a figure to the manuscript to depict the images used to do this said scoring. In its current form this manuscript does not provide a visual impression of the assessment being made to our readers.

Response: 

1. We have added the visual impression of the T-SGAS in the manuscript’s method section as figure 1. In our earlier referenced paper - “Patel A, Kurhe K, Prakash A, Bhargav S, Parepalli S, Fogleman E et al. Development and Validation of a Tablet-based Simplified Gestational Age Score (T-SGAS) for Early Identification of Preterm Neonates at Birth when Ultrasound Gestational Age Dating is Not Available: A Global Network for Maternal and Child Health Research Protocol. Available at: http://preprints.jmir.org/preprint/11913 DOI: 10.2196/preprints.11913.” we have described in detail how this simplified score was adapted with pictures on a tablet. 

2. Was the GA assessment also done by established norms like Dubowitz or Ballard's methods by a trained professional? If yes, then how these assessments varied compared to the app? If not then why not? Kindly explain it in the discussion part of the manuscript.

Response: 

The Simplified Gestational Age Score development on the tablet underwent the following two steps – 

Step - 1) We evaluated the 12-item New Ballard Score (NBS), 22-item Dubowitz score (DWS) and 11-item Meharban Singh (MS) and assessed the correlation of the total of these 3 scores and each item in these 3 scores with the Best obstetrics Estimate (BOE) (LMP and USG as gold standard). (Ref. Patel A, Lothe A, Belekar N, Thakur H. Development and initial validation of a simplified gestational age score in low birth weight newborns in India. Indian Journal of Child Health 2017;5(1): 15-19.) Through this exercise, we developed a 4-item simplified gestational age score (SGAS) for use in low birth weight newborns in India. SGAS showed promising results for accurate estimation of GA in LBW newborn and ease of use by nursing staff in the community.

Step – 2) We adapted the 4-item simplified gestational age score (SGAS) to a pictorial tablet app. This method was described in our referenced paper - “Patel A, Kurhe K, Prakash A, Bhargav S, Parepalli S, Fogleman E et al. Development and Validation of a Tablet-based Simplified Gestational Age Score (T-SGAS) for Early Identification of Preterm Neonates at Birth when Ultrasound Gestational Age Dating is Not Available: A Global Network for Maternal and Child Health Research Protocol. Available at: http://preprints.jmir.org/preprint/11913 DOI: 10.2196/preprints.11913.”

In the current study the ANMs were trained to use only the T-SGAs. The aim of this study was to assess if ANMs who otherwise have minimal training in assessment of gestational age are able to use the T-SGAS reliably to assess GA of newborns at birth. It was validated against the gold standard of GA at birth assessed using prenatal USG, which is also the gold standard for validating any newborn gestatational age assessment methods like NBS or any other scoring method. So validation against NBS or DWS was not considered for two reasons – 1) Any neonatal GA assessment score is a less accurate method for assessment of GA than prenatal USG, 2) Any additional training provided to ANMs on NBA or DWS would bias the evaluation skills of ANMs. 

This has now been explained in the discussion section.

3. Discussion needs to elaborate on how this tool can be used by ANM's and in what way it can be helpful to these designated infants <37 weeks?

Response: 

We have explained in the methods and discussion section that the ANM has to simply select the picture on the Tablet App that matches the what they observed in the newborn baby. The App then auto-calculated the score which categorizes the newborn as preterm < 37 weeks. The T-SGAS had high reliability and high enough likelihood ratios to warrant its routine use by ANMs to refer preterm newborns to higher facilties. 

4. Table 3 and 4 though helpful to a trained professional with good knowledge of biostatistics, does not help our average reader of PLOSone so should be either deleted or published as supplementary tables on the web.

Response: 

We have now moved table 3 and 4 to supplement. It is now S6a and S7a respectively 

Minor concerns:

1. Remove repetitive statements in introduction, methods and results to reduce the size of manuscript.

Response:

We have done as suggested.

2. Elaborate on educational status on ANM's in your methods section and why they were chosen to do the measurements, were they randomly selected or had special qualifications?

Response: 

 The educational level of ANMs is a diploma in midwifery of 2 years provided after high school education. The reason behind selecting, ANM’s for this study was that they provide midwifery service at all rural primary health centers. They were selected on random basis and trained in use of the T-SGAS App 

Response: 

 We have done the needful.

2. Please provide additional details regarding participant consent. In the Methods section, please ensure that you have specified what type of consent you obtained (for instance, written or verbal) and whether the ethics committee approved this consent procedure. If verbal consent was obtained please state why it was not possible to obtain written consent and how verbal consent was recorded. If your study included minors, state whether you obtained consent from parents or guardians.

Response: 

We have now provided the details in the methods section. 

 Response: 

ORCID iD-0000-0002-25587421 (Dr. Archana Patel) It has been validated in the editorial manager. 

Response: 

Indian Council of Medical Research does not allow data sharing unless the protocol for data sharing was previously approved by them, the Institutional Ethics Committee and consented by the participants. If data is requested by the journal for verification of results we will seek permissions from the Institutional Ethics Committee (LMRF-IEC), our Sponsors and the Indian Council of Medical Research. 

Contact details for LMRF-IEC:

Dr. Prabir Kumar Das, Member Secretary

Email ID- prabir_das23@rediffmail.com

However, other researchers may submit data access requests to Dr. Prabir Kumar Das, Member Secretary at LMRF-IEC (prabir_das23@rediffmail.com).”

 Response: 

Please see our response above. The contact information for data access our – 

 LMRF-IEC:

Dr. Prabir Kumar Das

Member Secretary

Lata Medical Research Foundation, 

Kinkini Kutir, Vasantnagar, Nagpur – 440022

Ph. No. 91-8805023450

Email.ID- prabir_das23@rediffmail.com

Response: 

Please see previous response.

"The authors declare no competing interests."

We note that one or more of the authors are employed by a commercial company: M&H Research, LLC

Response: 

While Dr. Kulkarni is affiliated to M&H Research, LLC, San Antonio, TX, USA; this work was done under the aegis of Lata Medical Research Foundation, Nagpur, India of which Dr. Kulkarni is the President and provides his services gratis. No funding was received from M&H Research, LLC for work reported here.

“The funder provided support to the institute for conduct of the study, but did not have any additional role in the study design, data collection and analysis, decision to publish, or preparation of the manuscript. The specific roles of these authors are articulated in the ‘author contributions’ section.”

Response:

The commercial affiliation had no role to play in our study.

 Response:

We have provided an explanation regarding the data sharing requirement. We shall remove the phrase. 

7. Your ethics statement must appear in the Methods section of your manuscript. If your ethics statement is written in any section besides the Methods, please move it to the Methods section and delete it from any other section. Please also ensure that your ethics statement is included in your manuscript, as the ethics section of your online submission will not be published alongside your manuscript.

Response: 

This has been updated in the method section.

Response: 

This has now been provided. 

Reviewers' comments:

Reviewer's Responses to Questions

Comments to the Author

1. Is the manuscript technically sound, and do the data support the conclusions?

Reviewer #1: Yes

2. Has the statistical analysis been performed appropriately and rigorously? 

Reviewer #1: Yes

3. Have the authors made all data underlying the findings in their manuscript fully available?

Reviewer #1: Yes

4. Is the manuscript presented in an intelligible fashion and written in standard English?

Reviewer #1: Yes

5. Review Comments to the Author

Reviewer #1: This is a cross-sectional study evaluating the accuracy of Tablet based gestational age estimation app in low resource setting. They have identified score <14 as good predictor of identifying preterm birth (<37 weeks) and 9% probability of preterm birth if score>13 and weight <2.5kg.

The app has high Inter & intra-observer variation and Likelihood ratio, though sensitivity is low. The study is well designed and the analysis is robust and potential for being a useful tool in low-resource settings.

Concerns:

1. 37 weeks was used as the cutoff for identifying preterm birth or not. How valid is the tool at other gestational ages - <32 weeks, <35 weeks etc. <37 weeks is important for epidemiological reasons to identify preterm birth. However, infants >35 weeks generally don’t require admission to NICU. This data would be more helpful for clinicians whether the baby needs to be transported to the NICU or not, which in a low resource NICU could be a significant cost.

Response: 

This is an interesting suggestion by the Reviewer. However, the suggested analyses are outside the scope of the current manuscript. We do nothave data on how many of the newborn babies needed to admitted to NICU since the study was conducted in primary health care settings. However, we do understand that our tool can have further improved implication if we do additional studies on the lines of the Reviewer’s suggestion. We have now added this as an implication in the Discussion section. 

2. Can you clarify if the ANM assessor’s correlation coefficient was for assessors from all three study sites?

Response: 

This is correct.

3. Likelihood ratios should be included in the abstract.

Response: 

This has been updated in the abstract.

---

## [Decision Letter · Decision Letter 1]

14 Aug 2020

Early identification of preterm neonates at birth with a Tablet App for the Simplified Gestational Age Score (T-SGAS) when ultrasound gestational age dating is unavailable: A validation study

PONE-D-20-09295R1

Dear Dr. Patel,

We’re pleased to inform you that your manuscript has been judged scientifically suitable for publication and will be formally accepted for publication once it meets all outstanding technical requirements.

Regards,

Prem Shekhawat, MD

Academic Editor

PLOS ONE

Additional Editor Comments (optional):

Reviewers' comments:

Reviewer's Responses to Questions

**Comments to the Author**

1. If the authors have adequately addressed your comments raised in a previous round of review and you feel that this manuscript is now acceptable for publication, you may indicate that here to bypass the “Comments to the Author” section, enter your conflict of interest statement in the “Confidential to Editor” section, and submit your "Accept" recommendation.

Reviewer #1: All comments have been addressed and submission is suitable for publication.

2. Is the manuscript technically sound, and do the data support the conclusions?

Reviewer #1: Yes

3. Has the statistical analysis been performed appropriately and rigorously? 

Reviewer #1: Yes

4. Have the authors made all data underlying the findings in their manuscript fully available?

Reviewer #1: Yes

5. Is the manuscript presented in an intelligible fashion and written in standard English?

Reviewer #1: Yes

6. Review Comments to the Author

Reviewer #1: All my concerns have been appropriately addressed. I hope the authors do plan to develop this project further and collect data on clinical outcomes, as the true use of the app would be then realized and widespread use can be advocated.

7. PLOS authors have the option to publish the peer review history of their article (what does this mean?). If published, this will include your full peer review and any attached files.

Reviewer #1: No

---

## [Editor Report · Acceptance letter]

18 Aug 2020

PONE-D-20-09295R1 

Early identification of preterm neonates at birth with a Tablet App for the Simplified Gestational Age Score (T-SGAS) when ultrasound gestational age dating is unavailable: A validation study 

Dear Dr. Patel:

I'm pleased to inform you that your manuscript has been deemed suitable for publication in PLOS ONE. Congratulations! Your manuscript is now with our production department. 

Kind regards, 

on behalf of

Dr. Prem Singh Shekhawat 

Academic Editor

PLOS ONE